# DNB-AI-Project at the GermEval-2025 LLMs4Subjects Task: KIFSPrompt - Knowledge-Injected Few-Shot Prompting

**Maximilian Kähler** and **Katja Konermann** and **Lisa Kluge**

Deutsche Nationalbibliothek

Leipzig/ Frankfurt am Main, Germany

**Correspondence:** m.kaehler@dnb.de

## Abstract

This work-in-progress report presents our system, KIFSPrompt, developed for the second phase of the shared task LLMs4Subjects at GermEval'25. The primary focus of this phase is the development of energy- and compute-efficient large language model (LLM) systems for subject tagging in library cataloging systems. A key challenge in this task is the requirement to select keywords from a large, normed vocabulary, the Integrated Authority File (GND). Building on our previous work ([12] and [13]), our system leverages few-shot prompting, where a generative LLM is presented with a limited number of examples of texts annotated with subject terms, and prompted to identify the most relevant subject terms for a new input text. To ensure alignment with the library's normed subject terms, we employ a mapping approach based on embedding similarity. We extend our previous work by incorporating a retrieval stage, which selects relevant few-shot examples from the training set to create knowledge-injected prompts, enabling the LLM to provide more specific and accurate keyword suggestions. This enhancement allows the LLM to adapt to the input text, resulting in improved performance in a single prompt.

## 1 Motivation

LLMs4Subjects [6, 7] is a shared task organized by TIB[1] aiming at utilising large language models (LLMs) for the task of automated subject indexing. In the previous round for LLMs4Subjects, the DNB-AI-Project submitted a prototype that employed a multitude of LLMs in a Few-Shot-Prompting ensemble to solve the subject tagging problem. While results did not outperform other methods when directly compared to the gold-standard subject terms that were intellectually annotated following the German rules for subject headings (RSKW[2]), our approach was ranked first in the qualitative ratings conducted by TIB's subject librarians. However, a major drawback of our previous work is the computational complexity involved in generating keywords from multiple LLMs with multiple prompts. During keyword generation, each test document would undergo a series of 20 LLM-calls with LLMs of varying size. With our on-premise hardware, generating subject terms for the LLMs4Subjects test set took 234 GPUh in total. While showing an increase in quality, this immense resource consumption disqualifies our previous system from being of practical relevance for (usually poorly equipped) library systems. In line with the main theme of developing compute- and energy-efficient systems in this second phase of the shared task, we aim at reducing the cost of the system by restricting ourselves to two LLM-calls per document: (*complete* and *rank*) complemented by two similarity search stages (*retrieve* and *map*).

Inspired by the work on analogical reasoning of LA²I²F [21] in the first round of LLMs4Subjects, our main advance in this second round is to add a similar retrieval step to our system pipeline. For any test document, the retrieval stage extracts the top $L$ documents from the training set most similar to the test document. The LLM prompt is then assembled from these most similar training examples, along with their annotated GND labels. This dynamic assembling of few-shot examples directly injects knowledge of related documents and the relevant GND subject terms into the prompt, bridging the problem that a pre-trained LLM would have no knowledge of relevant subject terms in the GND.

## 2 Related Work

One established perspective on subject tagging with a normed vocabulary is to approach the prob-

---

[1]TIB: German National Library of Science and Technology

[2]Regeln für die Schlagwortkatalogisierung, cf. https://d-nb.info/1126513032/34

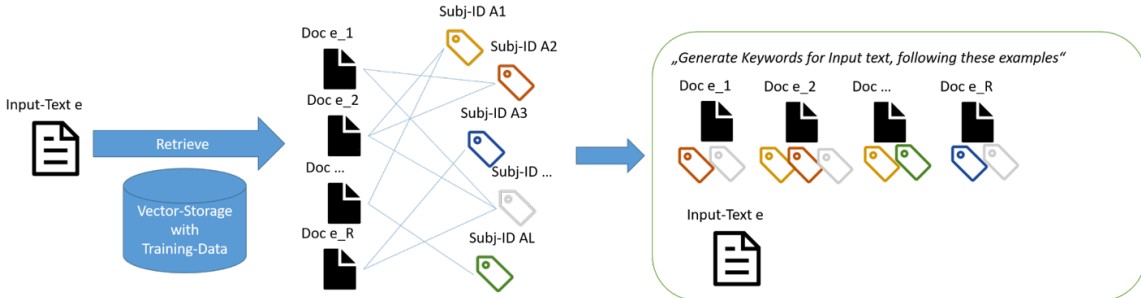

Figure 1: Assembling the knowledge injected few-shot prompt from training data. Training data is processed with an embedding model and indexed in a vector storage. Hybrid search enables retrieval of documents similar to the input document. A few-shot prompt is dynamically assembled using the retrieved training documents and their known subject terms.

lem as an extreme multilabel classification problem (XMLC). Dasgupta et al. [5] gives a recent overview of developments in XMLC algorithms. Using backends of the Annif toolkit, Poley et al. [19] used a combination of XMLC algorithms and classical lexical matching [18, 9] to address the problem of extracting subject terms from the GND in a productive library system.

Pushing forward the capabilities of Annif, Suominen et al. [22] extended Annif with another XMLC backend [3, 24] and used LLMs for translation as well as generation of synthetic data to achieve top performance in the first phase of LLMs4Subjects.

Combining retrieval techniques with generative language models goes back to Lewis et al. [15]'s work on retrieval augmented generation (RAG). The first phase of LLMs4Subjects [6] has stimulated many contributions to the field, that follow RAG-like approaches. Besides the work of Salfinger et al. [21], which directly inspired this new adaptation of our system, team Tian et al. [23] used a multi-stage LLM approach, including data synthesis, knowledge distillation, fine-tuning, retrieval, and re-ranking, which achieved high marks in quantitative and qualitative scores of LLMs4Subjects.

## 3 System Description

### 3.1 Stage Design

Our system can be described as a sequential process of four stages. We start with an input text $e$:

**1. Retrieve:** Based on embedding similarity, a vector search among the training documents is performed, retrieving the top $L$ documents most similar to the input text $e$. Based on the language of the input text, which is recorded in each documents metadata, only training documents of the

same language are selected.

**2. Complete:** A prompt is assembled containing the retrieved top $L$ documents and their annotated subject terms as few-shot examples, as well as the input text. An LLM generates keyword suggestions from this prompt.

**3. Map:** Generated keywords are mapped to subject terms in the target vocabulary using hybrid search [2] based on embedding similarity and BM25 [20].

**4. Rank:** An LLM ranks the suggested subject terms according to their relevance to the input text.

Figure 1 illustrates how the few-shot prompt is assembled from similar training documents and the input text. The system instruction and an example prompt for the *complete* stage is displayed in Table 1. Note, the translation in the table is only for readability. We did not translate texts or subject terms in our system. In particular, English examples would be presented with German subject terms inside a prompt. However, we restricted the retrieved examples to fit the language of the input text using the language metadata of the record. The example demonstrates how the retrieval stage selects few-shot examples that are highly similar to the input text.

The stages *Map* and *Rank* work as in our previous system, cf. [13]. Instructions for the rank stage can be found in appendix A.1. The system does not involve any fine-tuning and works with off-the-shelf open-weight models from Hugging Face[3].

---

[3] https://huggingface.co/

| | German Prompt | English Translation |
|---|---|---|
| INSTRUCTION | `<\|im_start\|>`system
Dies ist eine Unterhaltung zwischen einem intelligenten,
hilfsbereitem KI-Assistenten und einem Nutzer.
Der Assistent antwortet mit Schlagwörtern auf den Text des Nutzers.
`<\|im_end\|>` | `<\|im_start\|>`system
This is a conversation between an intelligent, helpful AI
assistant and a user.
The assistant responds with keywords to the user's text.
`<\|im_end\|>` |
| RETRIEVED EXAMPLE 1 | `<\|im_start\|>`user
Gleitende Übergänge in den Ruhestand: Wirkung auf die
Erwerbsbeteiligung und das Humankapital Älterer in Deutschland …
`<\|im_end\|>`
`<\|im_start\|>`assistant
Altersteilzeitarbeit, Arbeitsangebot, Humankapital, Wirkungsanalyse
`<\|im_end\|>` | `<\|im_start\|>`user
Gradual transitions into retirement: Effects on labor force participation
and human capital of older people in Germany …
`<\|im_end\|>`
`<\|im_start\|>`assistant
Partial retirement, labor supply, human capital, impact analysis
`<\|im_end\|>` |
| RETRIEVED EXAMPLE 2 | `<\|im_start\|>`user
Eingliedern statt ausmustern: Möglichkeiten und Strategien zur
Sicherung der Erwerbstätigkeit älterer Arbeitnehmer …
`<\|im_end\|>`
`<\|im_start\|>`assistant
Älterer Arbeitnehmer, Erwerbsfähigkeit, Gesundheitspolitik,
Personalpolitik
`<\|im_end\|>` | `<\|im_start\|>`user
Integrate instead of discarding: Options and strategies to secure
employment of older workers …
`<\|im_end\|>`
`<\|im_start\|>`assistant
Older worker, employability, health policy, personnel policy
`<\|im_end\|>` |
| FURTHER RETRIEVED EXAMPLES | … | … |
| TEST-TEXT | `<\|im_start\|>`user
Arbeit im Alter: Zur Bedeutung bezahlter und unbezahlter
Tätigkeiten in der Lebensphase Ruhestand …
`<\|im_end\|>`
`<\|im_start\|>`assistant | `<\|im_start\|>`user
Work in old age: On the importance of paid and unpaid
activities in the retirement phase of life …
`<\|im_end\|>`
`<\|im_start\|>`assistant |
| GENERATED KEYWORDS | 'Ruhestand', 'Gesundheit', 'Arbeit', 'Geschlecht', 'Alter' | 'Retirement', 'Health', 'Work', 'Gender', 'Age' |

Table 1: Example for a dynamically created few-shot prompt used for the *complete* stage. Input text is displayed in green. Generated keywords are printed in red.

## 3.2 Implementation Details

As embedding model for the *Retrieve* and *Map* stages, we used Chen et al. [4]'s BGE-M3-embeddings, for their good benchmark results and multilingual capabilities. Our vector search is powered by a local Weaviate[4] vector storage, enabling efficient HNSW-Search [16] in the target vocabulary (*Map*) and the training data set (*Retrieve*). The hybrid search is configured with a weight-value of $\alpha = 0.7$ between HNSW-Search and BM25. The number of retrieved documents was set to $L = 8$.[5] However, if the concatenation of the examples in the few-shot prompt exceeded the total length of 15,000 tokens, the number of examples was automatically reduced accordingly. For the generation stage (*Complete*) we used Mistral AI's `Ministral-8B-Instruct-2410`[6]

instruction-tuned eight billion parameter model. The ranking was conducted with Dubey et al. [8]'s `Llama-3.1-8B-Instruct`[7] of similar size. The models were chosen as large as possible, such that each model would fit a single NVIDIA A100 GPU with $80\,\text{GB}$ GPU RAM at 16-bit floating point precision for the model weights and still allow for a prompt length of up to $15\,000$ tokens. To have some independence between *Complete* and *Rank* we selected the models to stem from distinct base models and model providers. All stage processes were synchronized in a data processing pipeline using DVC [14]. Calculations were performed with on-premise hardware consisting of a 2 x Intel(R) Xeon(R) Gold 6338T CPU @ 2.10GHz (48 cores) architecture equipped with two NVIDIA A100 GPU processors.

---

[4] https://weaviate.io/

[5] See section A.3 in the appendix for a sensitivity analysis on the choice of $L$

[6] https://huggingface.co/mistralai/Ministral-8B-Instruct-2410

[7] https://huggingface.co/meta-llama/Llama-3.1-8B-Instruct

## 4 Metrics

The shared task evaluates results according to standard information retrieval metrics: Precision, Recall, F1-Measure and NDCG. These are computed as document averages with varying limits $k = 5, 10, 15, 20$ on the system's ranked output. As our system produces varying numbers of predicted subject terms per record, we added another metric, called R-Precision [17]:

$$\text{R-Prec} := \frac{\text{tp}}{\min(\text{tp} + \text{fp}, \text{tp} + \text{fn})} \quad (1)$$

with the usual notion of fp for false positives, tp for true positives, and fn for false negatives. R-Precision does not penalize a system if it produces fewer than $k$ subject terms per record, provided that the gold standard also assigns fewer subject terms.

## 5 Results

### 5.1 Quantitative Results

When comparing suggested subject terms with intellectually annotated subject terms on the shared task's test set, we find the following results (Table 2). We restrict ourselves to the upper part of the ranking ($k = 5$), as our system does not usually produce more than 5 suggested subject terms.[8] Appendix A.4 analyzes our system's behaviour in terms of varying numbers of predicted subject terms per record. We hypothesize that our system is adapting the number of predicted subject terms per record from related training examples.

| Team | Prec@5 | Recall@5 | F1@5 | R-Prec@5 |
|------|--------|----------|------|----------|
| annif | 0.272 | 0.513 | 0.323 | 0.525 |
| DNB-AI-Project | 0.202 | 0.395 | 0.247 | 0.466 |
| ubffm | 0.059 | 0.128 | 0.119 | 0.074 |

Table 2: Quantitative evaluation leaderboard on *Test-Set* ($n = 27,998$). The table contains only a selection of the official results, restricted to $k = 5$.

Our system is placed second out of three in this quantitative evaluation, with a significant margin to both contending teams. Notably, recall@5 is much higher than precision@5, indicating that a limit of $k = 5$ presumably does not strike an ideal precision-recall balance, in terms of F1-Score.

### 5.2 Qualitative Results

Opposed to the binary relevance rating of section 5.1, where a suggested subject term is either

---

[8]Visit the official task site for the full leaderboard.

right or wrong according to the gold standard, the qualitative evaluation by TIB's subject experts assesses the correctness of suggestions, regardless of the gold standard that was actually assigned. Subject terms counted as false positives in the quantitative evaluation can be marked as correct in the qualitative rating. This evaluation is titled *Case 1*. Table 3 shows the qualitative leaderboard for Case 1. Ratings were conducted for a full ranking of theoretically $k = 20$ suggested subject terms per record. As our system is not designed to produce suggestions of arbitrary length, again, we only include the results at $k = 5$.

| Team | Prec@5 | Recall@5 | NDCG@5 | F1@5 |
|------|--------|----------|--------|------|
| Annif | 0.820 | 0.353 | 0.871 | 0.494 |
| DNB-AI-Project | 0.520 | 0.580 | 0.707 | 0.548 |
| ubffm | 0.256 | 0.180 | 0.273 | 0.211 |

Table 3: Qualitative leaderboard Case 1 ($n = 50$)

In a second scenario, *Case 2*, subject librarians rated the usefulness of suggested subjects for retrieval purposes, which is more strict. General subject terms that are technically correct (and might even occur in the gold standard), but are irrelevant for retrieval from a user perspective, are ignored in this scenario. Table 4 summarizes the results for Case 2.

| Team | Prec@5 | Recall@5 | NDCG@5 | F1@5 |
|------|--------|----------|--------|------|
| Annif | 0.604 | 0.428 | 0.712 | 0.501 |
| DNB-AI-Project | 0.392 | 0.522 | 0.617 | 0.448 |
| ubffm | 0.160 | 0.151 | 0.192 | 0.155 |

Table 4: Qualitative leaderboard Case 2 ($n = 50$)

In the Case 1 evaluation, KIFSPrompt achieves the best results in terms of F1@5 among the three teams. However, Team Annif still outperforms our system in terms of NDCG@5 and higher-rank metrics, as well as in the Case 2 scenario. In both qualitative evaluation scenarios KIFSPrompt scores highest in terms of recall@5. Further results containing scores by subject group can be found in the appendix A.5.

### 5.3 Error Analysis

In our development cycle on a small dev-set subsample, our `KIFSPrompt` system was able to perform on par with our previous `llm-ensemble` approach. However, transferring the results to the shared task's test set did not work equally well. Table 5 compares the performance of our two systems `llm-ensemble` and `KIFSPrompt` on our dev-sample as well as the test set. `KIFSPrompt`

has a large drop in recall@5 and also loses a significant amount of precision, whereas the `llm-ensemble` has a smaller performance deterioration. One reason for this behaviour could be a data distribution shift between the training set and the test set. The `llm-ensemble` does not draw any knowledge from the training set, so its performance does not depend on the training data distribution. However, `KIFSPrompt` relies on similarity of training and test data. The more similar the training documents are to the test documents, the more likely it is that `KIFSPrompt` will find correct subject terms in the examples.

| Name | Prec@5 | Recall@5 | F1@5 | R-Prec@5 |
|---|---|---|---|---|
| *Test-Set (n = 27,998)* | | | | |
| KIFSPrompt | 0.202 | 0.395 | 0.247 | 0.466 |
| llm-ensemble | 0.247 | 0.473 | 0.301 | 0.485 |
| *Dev-Set-Subsample (n = 1,000)* | | | | |
| KIFSPrompt | 0.237 | 0.467 | 0.311 | 0.517 |
| llm-ensemble | 0.247 | 0.500 | 0.328 | 0.510 |

Table 5: Comparison of system performance of our two approaches on Test and Dev sets.

Further system comparison of `KIFSPrompt` with our `llm-ensemble` can be found in the appendix section A.2.

### 5.4 Saved Resources

With the rapid development of the field, it is difficult to make an explicit comparison with our previous system. For example, we found that already an upgrade to the latest version of our inference engine vLLM[9] significantly improved speed in the generation steps. Also, there is a constant time-overhead for loading a model. Averaged over the test set of 27,998 documents, the completion stage for our llm-ensemble approach was at $30.1\,\mathrm{s}/it$, whereas our `KIFSPrompt` system is at $0.5\,\mathrm{s}/it$. Thus, our system uses less than two percent of the resources, in contrast to our previous approach. This saving in resources can be attributed to two facts: Firstly, during the completion step we only need one LLM call (in contrast to 20 previously). Secondly, we restrict to the smaller model `Ministral-8B-Instruct-2410`, rather than invoking large models such as `Meta-Llama-3.1-70B-Instruct` or `Mixtral-8x7B-Instruct-v0.1` in the LLM-Ensemble.

---

[9]version 0.6.4 $\rightarrow$ 0.9.0.1

## 6 Conclusion and Future Work

The introduction of Knowledge-Injected Few-Shot Prompting yields a significant reduction in costs compared to our previous approach, making it a viable option for resource-constrained institutions. Notably, `KIFSPrompt` can be seamlessly integrated with off-the-shelf language models, eliminating the need for costly training or fine-tuning. This makes it an attractive solution for libraries or institutions with limited hardware budgets. However, the optimal balance between quality and resources may vary depending on the specific use case and circumstances. The qualitative ranking shows, that our system can outperform even sophisticated ensemble approaches, such as Annif, in the $k <= 5$ part of the ranking, which is most relevant for practical library systems. As such, `KIFSPrompt` presents a valuable alternative for certain institutions, further expanding the quality-resource landscape of subject indexing systems and providing a useful benchmark for future developments. Using LLMs for subject indexing remains a promising road. There are ample opportunities to enhance such systems. Combining `KIFSPrompt` with our previous `llm-ensemble` approach could be a promising way to further improve quality. Additionally, evaluating `KIFSPrompt` with smaller language models may lead to further resource savings.

In the shared Task LLMs4Subjects, keywords are organized in a knowledge graph-like structure. Future research could explore how to leverage the relationships between keywords as an additional source of information for subject indexing using LLMs, cf. [11, 25, 1]. Moreover, fine-tuning a generative model specifically for this task may yield further performance improvements by adapting its parameters to the domain. Investigating parameter-efficient fine-tuning methods (cf. [10]) which reduce training time and memory consumption could also be advantageous in this context.

### Acknowledgements

We thank the anonymous reviewers for their helpful feedback. This work is a result of a research project at the German National Library (DNB)[10]. The project is funded by the German Minister of State for Culture and the Media as part of the national AI strategy.

---

[10]https://www.dnb.de/ki-projekt

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

# A Appendix

## A.1 Instructions for *rank*

Figure 2 shows the instructions used for the ranking stage, also used in our previous system. We found that best results are achieved by presenting keywords individually and not as a set of keywords for each input text.

> Du erhälst einen Text und ein Schlagwort. Bewerte auf einer Skala von 0 bis 10, wie gut das Schlagwort zu dem Text passt. Nenne keine Begründungen. Gib nur die Zahl zwischen 0 und 10 zurück.
> *You receive a text and a keyword. Rate on a scale from 0 to 10 how well the keyword fits the text. Do not provide any explanations. Return only the number between 0 and 10.*

Figure 2: Instruction used for the *rank* stage.

## A.2 System comparison KIFSPrompt vs. llm-ensemble

We include a precision-recall analysis on our dev-sample[11] ($n = 1,000$ documents), comparing `KIFSPrompt` with our previous `llm-ensemble` approach, cf. Figure 3. As a baseline, we also include the best results achieved with our single best hard-prompt (no dynamic few-shot examples) and best LLM (`Mixtral-8x7B-Instruct-v0.1`) from the previous study. All three curves include the LLM-ranking stage with `Llama-3.1-8B-Instruct` as post-processing. Table 6 shows point estimates for precision and recall that achieve optimal F1-score on the PR-curves. The optimal calibration is achieved by varying thresholds on confidence levels and limits (i.e. number of subject suggestions). We conclude that, with optimal calibration, `KIFSPrompt` can outperform the `llm-ensemble` in terms of optimal precision-recall balance, whereas the `llm-ensemble` achieves the best results in terms of PR-AUC.

| Ensemble Type | Precision | Recall | F1 | PR AUC |
|---|---|---|---|---|
| KIFSPrompt | 0.582 | 0.396 | 0.431 | 0.308 |
| llm-ensemble | 0.488 | 0.459 | 0.421 | 0.412 |
| one-model-one-prompt | 0.461 | 0.384 | 0.380 | 0.235 |

Table 6: Comparison of systems on dev-sample ($n = 1,000$). Point estimates for precision, recall and F1-score correspond to the F1-optimal calibration, displayed as cross-marks in figure 3.

---

[11]a subsample of the official LLMs4Subjects dev set

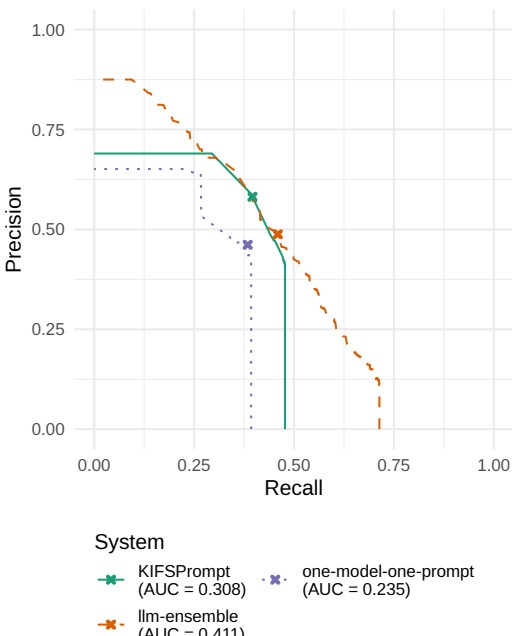

Figure 3: Precision-Recall curves for `KIFSPrompt`, `llm-ensemble` and baseline evaluated as doc-averages over our dev-sample. Cross-marks indicate the point of F1-optimal calibration for each curve, cf. Table 6 for point estimates.

## A.3 Sensitivity Analysis: Number of Retrieved Documents $L$

Investigating our system in the post-submission phase, we conducted a sensitivity analysis regarding the fluctuation of results with respect to the parameter $L$. Figure 4 shows F1@5 for various values of $L$. We see some variation in scores, however, there is no strong correlation with $L$.

## A.4 Distribution of Label-Numbers

A feature of `KIFSPrompt` seems to be its ability to learn the number of expected subject terms from the examples. On our dev set of $n = 1,000$ documents we compared the number of gold standard labels per document with the number of predicted labels and found that the dynamically selected examples seem to provide the LLM with a measure of how many labels to expect.

Figure 5 compares the distributions in the number of predicted subject terms.

The jitter plot in figure 6 demonstrates how KIFSPrompt produces varying numbers of predicted subject terms per Document.

The number of suggested subject terms may also depend on the sampling parameters for the LLM inference engine. Table 7 lists the settings for the

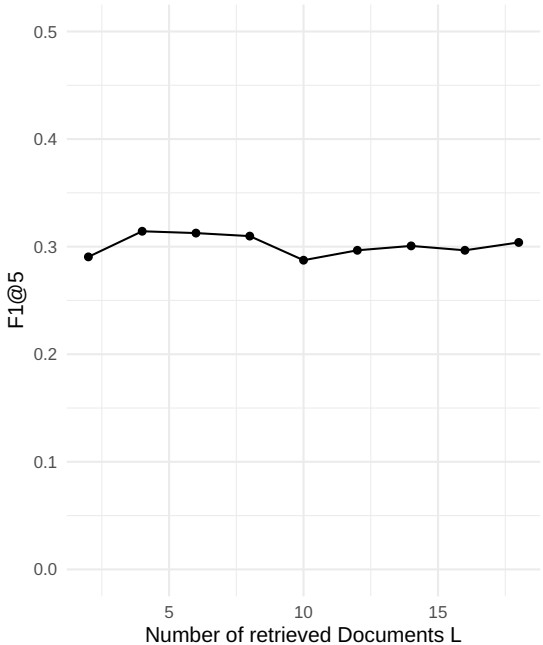

Figure 4: Sensitivity analysis: F1@5 as a function of the number of retrieved documents $L$ used in the few-shot prompt. Metrics are computed on dev-sample ($n = 1,000$).

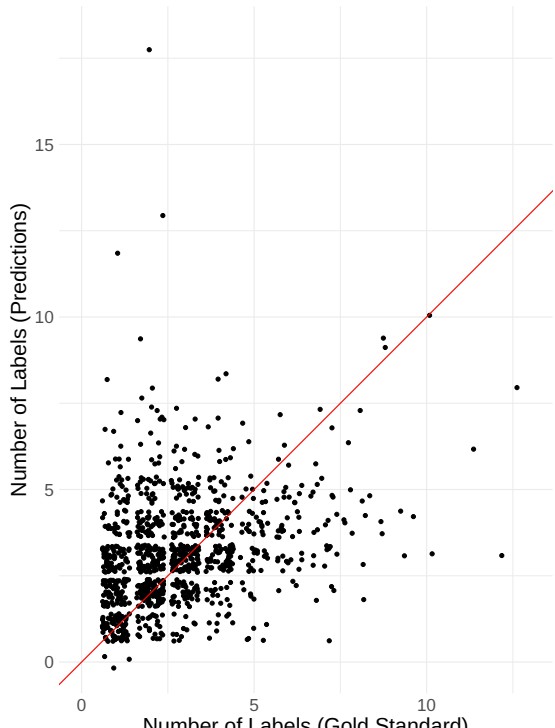

Figure 6: Jitter plot showing the number of predicted subject terms per document compared to the gold standard.

| Parameter | Value |
|---|---|
| max_tokens | 100 |
| min_tokens | 36 |
| temperature | 0 |
| presence_penalty | 0 |
| frequency_penalty | 0 |
| repetition_penalty | 1 |
| top_p | 1 |

Table 7: LLM sampling parameters for keyword generation using vLLM during the *Complete* stage.

In each subject group suggested subject terms for $n = 10$ records were evaluated by subject librarians of the TIB. KIFSPrompt performs best in economics and worst in literature studies.

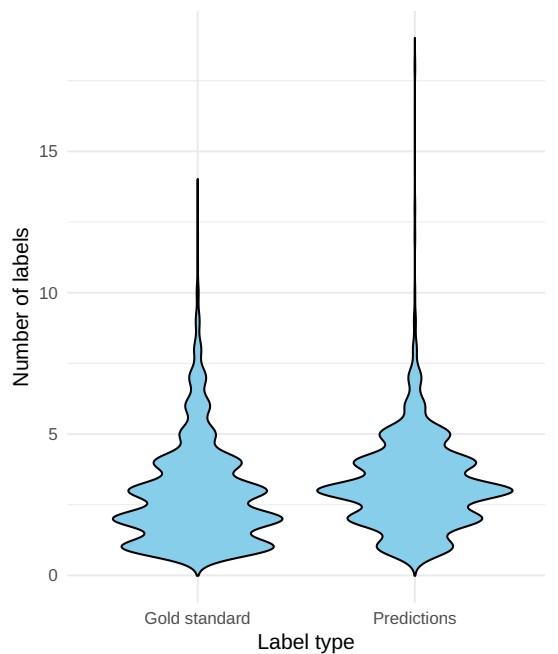

Figure 5: Distribution of the number of predicted subject terms per document compared to the gold standard, visualized as a violin plot.

sampling parameters used in our experiments.

## A.5 Qualitative Results per Subject Group

Figure 7 shows the qualitative results per subject group and team for evaluation scenario *Case 1*.

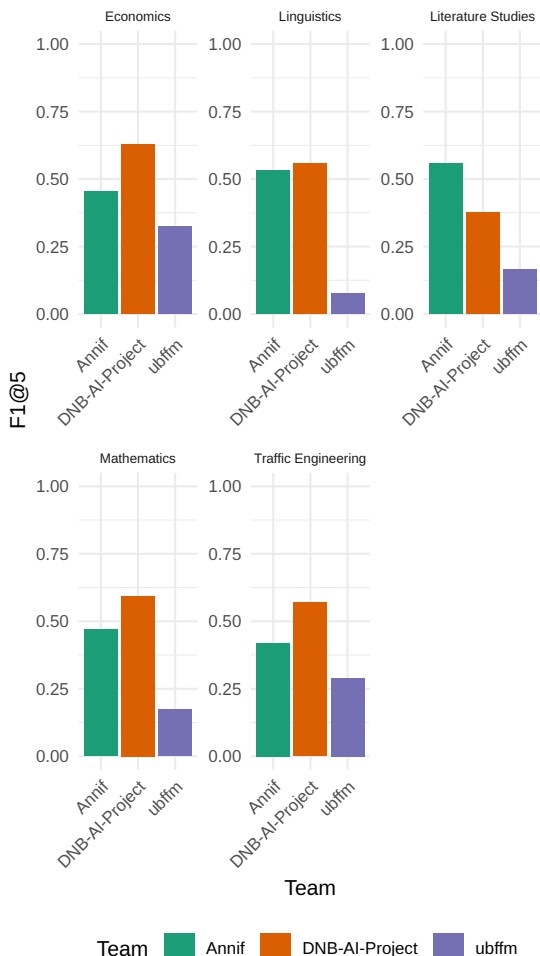

Figure 7: Qualitative F1@5 results per subject group.
Evaluation scenario *Case 1*.