# OpenReview forum: "DNB-AI-Project at the GermEval-2025 LLMs4Subjects Task: KIFSPrompt - Knowledge-Injected Few-Shot Prompting"
_GSCL.org/KONVENS/2025/Workshop/GermEval — GermEval25 Oral_

### Official Review · Reviewer_Jxst · 2025-08-14
**KIFSPrompt—Knowledge-Injected Few-Shot Prompting Review**

**Rating:** 4
**Confidence:** 3

**Review:**

Strengths:
- Well written, understanble and builds nicely on previous work.
- Their approach to knowledge injection effectively addresses the theme of the second phase of the shared task LLMs4Subjects at GermEval’25, as it eliminates the need to pre-train large language models (LLMs) on the large-scale Integrated Authority File (GND). Additionally, pretrained LLMs often struggle to capture the internal relationships present in GND data, and by adapting to the input text with relevant examples, the performance of the LLM is enhanced within a single prompt.

- The completion stage for the previous LLM-ensemble approach averaged “30.1 seconds per document,” whereas KIFSPrompt achieves this efficiency at “0.5 seconds per document.” They also used smaller, Off-the-Shelf Models, rather than larger models.

- They performed Best F1@5 among teams in case 1 (correctness regardless of gold labels) and had the highest recall@5 in case 2.

Weaknesses:

- The system typically does not produce more than 5 suggested subject terms per document and is, as mentioned in the paper, "not designed to produce suggestions of arbitrary length." It means KIFSPrompt is less suitable if a higher quantity of relevant subject terms (e.g., for k > 5) is desired or required.

- KIFSPrompt's performance is not uniform across all domains, performing best in economics and Mathematics and worst in literature studies among the evaluated subject groups. This could indicate a possible incompatibility in how effective it is based on the given subject matter being catalogued.

Recommendations for improvement:

- Please correct Figure 5’s alignments and write more detailed captions for images and tables.

- Table 1 is not mentioned in the text.

- Add the correct citation as mentioned in the email for “{S}em{E}val-2025 Task 5: {LLM}s4{S}ubjects - {LLM}-based Automated Subject Tagging for a National Technical Library{'}s Open-Access Catalog"

- The "Related Work" section could improve its writing style; it feels disconnected, jumping between methods without smooth transitions. - I couldn’t quite understand the purpose of “A2.1 Distribution of Label Numbers.”

- Please discuss potential dataset biases in TIBKAT. As you mentioned, “KIFSPrompt's performance relies on the similarity of training and test data; if training documents are not sufficiently similar to test documents, it is less likely to find correct subject terms from the few-shot examples.” Is there a proposed solution to address this issue?

- Please add the prompt sample you used for generation and ranking. For example, provide a brief illustration that includes an input text, several retrieved few-shot examples, and the generated subject terms to help readers visualize the pipeline.

- The choice of the top L retrieved documents is important. Please show results for different L values to justify the setting.

Comment for future work:

- Maybe try parameter-efficient fine-tuning methods on smaller LLMs to bridge the training–test distribution gap.

**Summary:**

The paper presents a novel method called KIFSPrompt (Knowledge-Injected Few-Shot Prompting) for subject indexing, which creates an energy- and compute-efficient system by restricting it to two LLM calls per document—for completion and ranking—followed by two similarity search stages—retrieve and map—aligning with the task's theme of efficiency. It features a well-designed multi-step pipeline that dynamically injects knowledge into the LLM prompt by selecting the most similar training documents and their annotated GND (Integrated Authority File) labels as few-shot examples, thereby enabling the LLM to provide more accurate keyword suggestions despite not being pre-trained on the GND labels.

---

### Official Review · Reviewer_d2pL · 2025-08-14
**Good attempt at creating an innovative compute-efficient LLM-based system for automated subject indexing.**

**Rating:** 4
**Confidence:** 5

**Review:**

Strengths:

- Well written, concise and clear
- Builds nicely on previous work, with a good Related Work section
- Innovative, original and interesting approach, clearly an useful contribution to the field, and matches well the theme of resource-efficient automated subject indexing systems
- Decent amount of error analysis and future ideas
- The system shows a good trade-off between output quality and computational efficiency. It is much more resource-efficient than the previous system while still delivering relatively good quality results, especially for the development set.

Weaknesses:

- Relies a lot on the previous SemEval paper. The reader is expected to be familiar with the previous system, or refer to the older paper. Does not work well as a stand-alone description, because important details about the system (e.g. Map and Rank) are omitted.
- There is no comparison to baseline methods, nor any testing of different methods or variations except for the comparison to the earlier system.

Questions to the authors

- Section 3.2: "The models were chosen as large as possible, such that each model would fit a single NVIDIA A100 GPU." This is confusing. First, A100 variants can have either 40GB or 80GB VRAM, which one did you use? Even without quantizing, larger models than 8B should fit on an A100 (roughly 12-15B for 40GB, or 24-30B for 80GB, depending in part on context size, inference engine and settings.) With quantizing, much larger models are possible.
- The recall@5 values for the qualitative evaluation (both case1 and case2) for your system were quite high, much higher than your precision@5. I don't understand how this could happen, and it differs from the two other systems where precision@5 > recall@5, as one would expect in this setting. Do you have an explanation? Is it possible that there is a mistake in the qualitative evaluation metrics? (Also, on the official qualitative leaderboard, your recall@K values are always the same regardless of K, which I find surprising.)

Possible mistakes, typos etc:

- Section 5.4: "Saved ressources" -> "Saved resources"

**Summary:**

The paper describes a system  for automated subject indexing that targets subtask 2 of the LLMs4Subjects shared task. It implements a four-stage process for determining relevant GND subject terms for a given document. The Retrieve and Map stages use an embedding model and vector search, while Complete and Rank stages use off the shelf LLMs. The system is partly based on an earlier system from the previous LLMs4Subjects round, but it is much more streamlined and uses a fraction of the resources. The result quality is quite good especially on the development set. The system was ranked 2nd out of three participating systems in the quantitative and qualitative evaluations.

---

### Decision · Program_Chairs · 2025-08-14

Accept (Oral)